# The Application of Balance Exercise Using Virtual Reality for Rehabilitation

**DOI:** 10.3390/healthcare10040680

**Published:** 2022-04-04

**Authors:** Yukio Urabe, Kazuki Fukui, Keita Harada, Tsubasa Tashiro, Makoto Komiya, Noriaki Maeda

**Affiliations:** Graduate School of Biomedical and Health Sciences, Hiroshima University, Hiroshima 734-8553, Japan; kazuki-fukui@hiroshima-u.ac.jp (K.F.); keitan-harada@hiroshima-u.ac.jp (K.H.); tsubasatashiro716@hiroshima-u.ac.jp (T.T.); makoto-komiya@hiroshima-u.ac.jp (M.K.); norimmi@hiroshima-u.ac.jp (N.M.)

**Keywords:** rehabilitation, virtual reality, balance training, center of pressure

## Abstract

To prevent falls, it is important to devise a safe balance training program that can be easily performed. This study investigated whether tilting an image in virtual reality (VR) can generate a center-of-gravity sway. Five men and five women were asked to rest standing upright (control condition) and to rest standing upright with a head-mounted display showing a tilted virtual image (VR condition), and changes in their standing balance were observed. Standing balance was assessed by measuring the distance traveled by the center of pressure (COP) of each of the participants’ legs. In order to investigate the effects of different tilt speeds and angles on COP, four different images were displayed in VR: an image tilting to 10° moving at a rate of 1°/s; an image tilting to 20° moving 1°/s; an image tilting to 10° moving 10°/s; an image tilting to 20° moving 10°/s. Change in COP was significantly greater in the VR than in the control condition (*p* < 0.01), and a tilt of 10° moving 1°/s showed the greatest change in COP (*p* < 0.01). Tilting an image in VR while in a resting standing position can change an individual’s COP; thus, VR may be applied to balance training.

## 1. Introduction

Trauma and fractures caused by falls can significantly reduce the activities of daily living of older adults, making their prevention a global issue [1]. One important factor that helps in the prevention of falls is the improvement of balance ability [2]. Conventional exercises using balance discs and balance pads have previously been shown to improve balance ability. Exercising with these tools activates the postural control response to maintain the body’s center of gravity within the supporting basal plane on an unstable surface that tilts back and forth, and left and right. This in turn improves the body’s ability to balance against external disturbances [3]. This is further challenged with greater center of gravity sway, which makes balancing even more difficult [4]. However, in reality, for many older adults it is too hard to maintain a resting standing position, and for some, balance exercises themselves may be too difficult. Thus, in this study, we propose a new method of balance training for older adults.

One way to induce center of gravity sway without performing balance exercises is to use virtual reality (VR) technology [5,6,7]. VR technology uses a head-mounted display (HMD) to show 360° images and visually place the user in a virtual space. Therefore, we thought that it would be possible to induce center of gravity sway by having the user experience a state in which their body is tilted and out of balance in this virtual space, similar to that in balance exercises using conventional balance tools [8]. If the gravity center sway is induced by only tilting the VR image, this practice can be applied to balance training with less difficulty. However, it is unclear to what extent the inclination of the image in the VR will cause the center of gravity to sway in an individual standing in a resting position.

Therefore, the purpose of this study was to investigate whether changes in the center of pressure (COP), which is considered an index of standing balance, can be induced only by the tilt of VR images.

## 2. Materials and Methods

### 2.1. Participants

Five men and five women were included in the study to measure COP mobility in both legs. The mean participant age was 21.7 years (SD = 0.9). The mean participant height was 164.9 cm (SD = 11.0), and the mean weight was 58.1 kg (SD = 10.2). All participants recruited for this study were students enrolled in Hiroshima University. The inclusion criteria were being 20 years old or older and having no experience using VR. The exclusion criteria were having orthopedic diseases of the lower limbs or VR sickness after wearing VR. This study was conducted in accordance with the guidelines of the Declaration of Helsinki and approved by the Epidemiology Ethics Committee of Hiroshima University (approval ID: E-2299). All the participants provided written informed consent before participating in the study.

The sample size required for the one-way repeated measures (ANOVA) test (effect size = 0.40 [large], α error = 0.05, power = 0.80) was calculated using the G* power 3.1 [9]. Using this test, we determined a minimum of 10 participants was required for this study.

### 2.2. Measurement Methods and VR Images

Measurements were taken under two conditions: one in which the participant held a resting standing posture with eyes open without wearing the HMD (control condition), and the other in which the participant was in a resting standing posture while viewing VR images on an HMD (VR condition; Figure 1).

For the tilt of the VR images, we used images of the laboratory landscape taken beforehand with a 360° camera (Key Mission 360, Nikon). In the VR condition, to investigate the differences due to speed and tilt, we used the following images: 1°/s, tilted 10° (VR1), 1°/s, tilted 20° (VR2), 10°/s and tilted 10° (VR3), and 10°/s and tilted 20° (VR4; Figure 2).

### 2.3. Assessment of COP

The participant placed both of their feet shoulder width apart and both of their arms to their sides (Figure 1). They held this standing position on a center-of-gravity sway meter (T.K.K. 5810, Takei Instruments) placed on an inclined table. 

In the control condition, the distance travelled was calculated from the movement trajectory of the COP for 10 s (Figure 3). The COP movement distance is defined as the linear distance between the COP coordinates at the beginning and end of the tilt. Three trials were conducted for each condition, and the average value was taken as the representative value. 

To investigate the amount of COP movement due to differences in the speed and tilt angle of the VR images, the participants were in the standing position on the center of gravity sway meter for 10 s in the VR1 condition, 20 s in the VR2 condition, 1 s in the VR3 condition, and 2 s in the VR4 condition. For each subject, the VR conditions were performed in random order, randomly assigned by random numbers using a computer.

### 2.4. Statistical Analysis

SPSS software (version 27.0; SPSS Japan Inc., Tokyo, Japan) was used for the statistical analysis. The normality of all variables was confirmed using the Shapiro–Wilk test. A paired-samples *t*-test was used to compare the control and VR1 conditions. Repeated measure ANOVA was then conducted among the four conditions of VR1, VR2, VR3, and VR4 to compare the differences based on the speed and tilt angles of the VR images. A Bonferroni test was used for post-hoc testing. The significance level was set at 5%. Values are presented as the mean ± standard deviation.

## 3. Results

The mean COP movement distance was 2.2 ± 0.9 mm in the control condition and 9.3 ± 3.6 mm in the VR1 condition, with significantly higher values in the VR1 condition (*p* < 0.01). The mean COP movement distances for the different VR images are listed in Table 1. The results of the multiple comparison test showed that VR1 was significantly higher than VR2, VR3, and VR4 (*p* < 0.01).

## 4. Discussion

In the current study, we investigated whether an individual’s COP is impacted by tilting the floor surface in a VR image. The results show that regardless of gender, COP is affected when an individual experiences inclination in the virtual space, without truly experiencing the incline. Furthermore, this is the first study to show that a 10° incline moving at a rate of 1°/s has the greatest influence on COP in comparison to other variations.

We consider that the amount of COP movement in the control condition (2.2 ± 0.9 mm) was determined to be valid without significant error compared to a previous study in which COP movement was measured in healthy subjects (1.4 ± 0.3 mm) [10]. The present study then showed that participants’ COP shifted more in the VR condition than in the control condition, indicating that COP shifts when an image is tilted, even in an individual is in a resting upright position. This phenomenon is known as vection [11]. Vection is a phenomenon in which an object that is actually stationary is perceived as moving because of changes in visual information [12,13]. This effect is greater when vision changes occur across the entire field of view [14,15], as in the VR condition using an HMD in this study. It is thought that an individual’s COP moves in this case because they visually perceive that they are tilted. Previous studies have reported that the directional information regarding one’s entire environment (e.g., the ceiling and floor) affects their inclination of their own body [16].

Further, previous studies have reported that the vection effect depends on the speed of the visual stimulus; that is, the faster the video, the larger the vection effect [17,18]. However, in the present study, the VR1 condition (1°/s) was shown to be the most effective in terms of the optimal speed and tilt angle needed to induce vection. With respect to velocity, it has been reported that the center of the retina has superior sensitivity to low velocity and the periphery to high velocity [19]. Furthermore, when considered in terms of frequency, it has been shown that lower frequencies (below 1 Hz) result in greater COP migration [20]. In a study on the perception of gravity by visual stimuli, it was reported that the vection effect does not increase any further when the tilt angle exceeds 20 degrees, but rather diminishes [21]. These factors may have caused the VR1 condition (with a tilt angle of up to 10 degrees at 1°/s) to have the strongest vection effect, resulting in a greater degree of COP movement.

These results suggest that simply having participants wear HMDs and watching slow images in VR could be applied to balance training. The intervention using VR images might be effective in improving the balance ability of elderly people who have difficulty in controlling their posture beyond the resting position.

One of the limitations of this study was that the participants were healthy university students. It is unclear whether the results of this study can be directly applied to older adults with impaired balance abilities. However, since it has been previously reported that older adults are susceptible to the vection effect [22], and since it has been reported that stability in a state using virtual reality is located between the open and closed eye states [20], the present results may have potential for application. In addition, we did not measure whether participants’ balance abilities changed before or after viewing the inclination of the VR image. In the future, it will be necessary to verify the effects of balance training using VR images.

## 5. Conclusions

In this study, we examined whether the tilt of the image in VR affects the sway of the COP in young adults. Our results show that individuals’ COPs changes more when an image is tilted in VR than when standing at rest. Furthermore, the results of a comparison of conditions in which the tilt angle and tilt speed of the VR were varied showed that the COP movement distance was significantly greater in the VR1 condition with a 10° tilt at 1°/s than in the VR2 condition with a 20° tilt at 1°/s, the VR3 condition with a 10° tilt at 10°/s, and the VR4 condition with a 20° tilt at 10°/s. This suggests that simply tilting the floor surface of the VR image could be used as a way to assist in balance training. Moreover, its clinical significance for the setting of conditions during VR balance training is presented.

## Figures and Tables

**Figure 1 healthcare-10-00680-f001:**
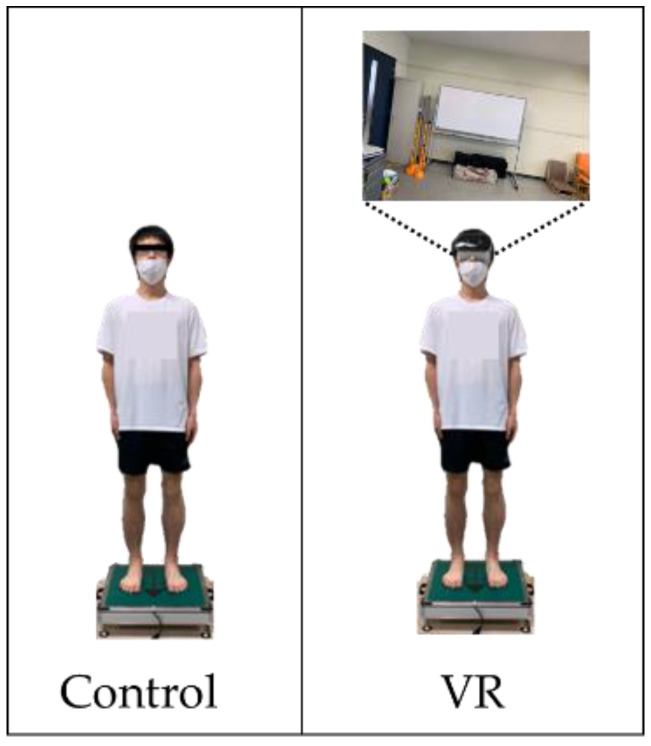
Center of pressure (COP) measurement condition ((**left**): control, (**right**): VR).

**Figure 2 healthcare-10-00680-f002:**
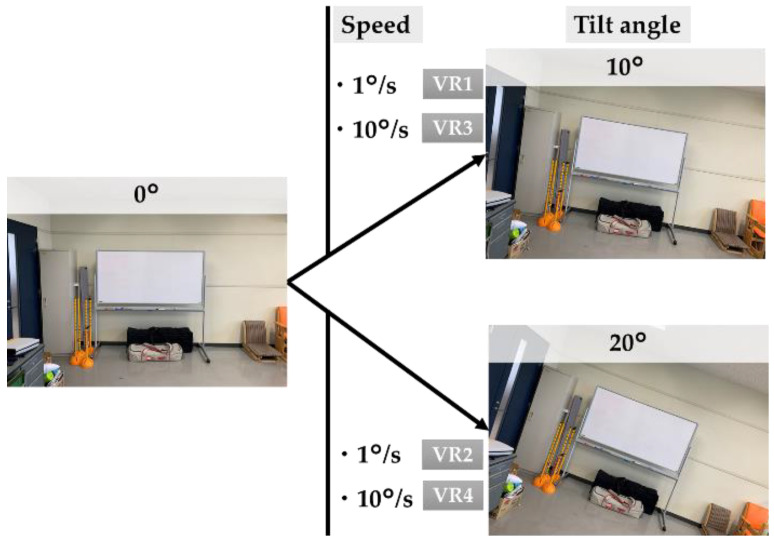
Content of the virtual reality (VR) videos.

**Figure 3 healthcare-10-00680-f003:**
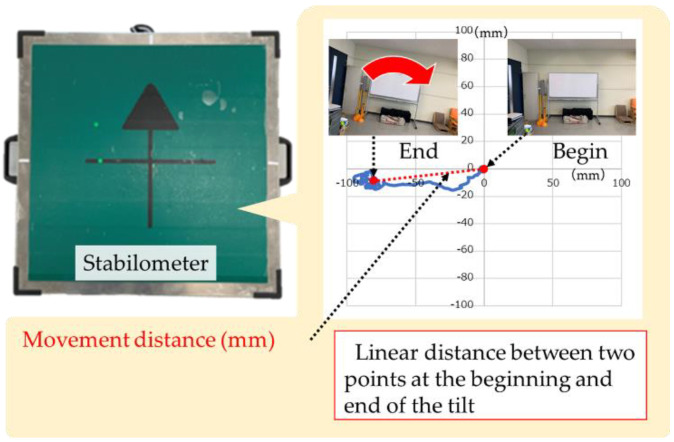
Method for measurement of center of pressure (COP) movement.

**Table 1 healthcare-10-00680-t001:** COP movement distance for four types of VR images.

Measurement	Condition				*F* Value	*p*-Value *
	VR1	VR2	VR3	VR4		
COP movement distance (mm)	9.2 ± 3.4 ^a,b,c^	5.4 ± 2.1	3.4 ± 0.9	4.3 ± 2.3	10.3	**<0.001**

COP = center of pressure; values are presented as mean ± standard deviation; * *p* < 0.05, considered significant difference (indicated with **bolded font**); ^a^ significant difference between the VR1 and VR2 (*p* = 0.009); ^b^ significant difference between the VR1 and VR3 (*p* < 0.001); ^c^ significant difference between the VR1 and VR4 (*p* = 0.001); VR2 and VR3, VR2 and VR4, and VR3 and VR4 are not significant difference.

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
