# Peer review of "The Application of Balance Exercise Using Virtual Reality for Rehabilitation"

_healthcare, 2022, doi:10.3390/healthcare10040680_

Round 1

Reviewer 1 Report

My first major remark concerns the concept of the research design. The authors studied the impact of different tilt speed and angle on centre-of-gravity sway for four different images displayed VR1, VR2, VR3, VR4. But, the authors didn't explain in what sequence they made it and if they tried to change it. Maybe, a 10° incline moving at a rate of 1°/s hasn't the greatest influence on COP in comparison to other variations if be done as not be the first test. So, I recommend doing this investigation in random order again. 
Moreover, the bibliography in the article should be extended and should be discussed with appropriate references. There is lacking citations of works that would complement the subject matter discussed e.g. publication DOI:https://doi.org/10.3390/s20247138.
In the aspect of “boundary condition” the rate of tilt speed is not only considered but also frequency, see e.g.  DOI:10.2478/hukin-2021-0004.
Please also compare your method of the control condition with DOI:10.5277/ABB-01082-2018-02.
The discussion and conclusion are too short. They should be improved and limitations should be addressed further.
I disagree with the authors' finding: "We found that the intervention using VR images was effective in improving the balance ability of elderly people who have difficulty in controlling their posture beyond the resting position". 
There is no proof to confirm it because only young people were involved in this investigation. In future, it will be necessary to verify the effects of balance training of elderly people using VR images.
I reconsider publishing this article after a major revision (extending quotations, correcting errors in the text, improving experiments as well as improving discussion and a conclusion).

Author Response

Response to Reviewer 1 Comments

Dear Editors:

Thank you for giving me the opportunity to submit a revised draft of my manuscript titled “The application of balance exercise using virtual reality for re-habilitation.” to Journal of Healthcare. We appreciate the time and effort that you and the reviewers have dedicated to providing your valuable feedback on my manuscript. We are grateful to the reviewers for their insightful comments on our paper. We have been able to incorporate changes to reflect most of the suggestions provided by the reviewers. We have highlighted the changes within the manuscript in yellow. Here is a point-by-point response to the reviewers’ comments and concerns indicated in red font.

Comments

My first major remark concerns the concept of the research design. The authors studied the impact of different tilt speed and angle on centre-of-gravity sway for four different images displayed VR1, VR2, VR3, VR4. But, the authors didn't explain in what sequence they made it and if they tried to change it. Maybe, a 10° incline moving at a rate of 1°/s hasn't the greatest influence on COP in comparison to other variations if be done as not be the first test. So, I recommend doing this investigation in random order again.

Response: Thank you for pointing this out. As you pointed out, it is possible that the results may vary depending on the order of measurement. In consideration of this point, the order of measurement was randomized. We have added this information to the manuscript because we did not include it. (p3, L94-95)

Moreover, the bibliography in the article should be extended and should be discussed with appropriate references. There is lacking citations of works that would complement the subject matter discussed e.g. publication DOI:https://doi.org/10.3390/s20247138. In the aspect of “boundary condition” the rate of tilt speed is not only considered but also frequency, see e.g.  DOI:10.2478/hukin-2021-0004.

Response: Thank you for your valuable suggestions. We have supplemented the discussion with references provided by you. (p4, L141-147)

Please also compare your method of the control condition with DOI:10.5277/ABB-01082-2018-02.

Response: Thank you for your valuable suggestions. Comparing the results of the control conditions in the literature you mentioned with the results of the present control conditions, there is no significant difference, and we believe that the measurement results are reasonable. We have added the following note to the manuscript. (p4, L125-127)

The discussion and conclusion are too short. They should be improved and limitations should be addressed further.

Response: Thank you for pointing this out. We have added our discussion and conclusions based on the references you provided. Furthermore, at the limitation, we have revised and added the content based on the effect of video stimulation using a head-mounted display on postural control.

I disagree with the authors' finding: "We found that the intervention using VR images was effective in improving the balance ability of elderly people who have difficulty in controlling their posture beyond the resting position".

There is no proof to confirm it because only young people were involved in this investigation. In future, it will be necessary to verify the effects of balance training of elderly people using VR images.

Response: Thank you for pointing this out. As you say, our wording may have been a leap, so we have revised it as follows and highlighted it in yellow on the manuscript. “The intervention using VR images might be effective in improving the balance ability of elderly people who have difficulty in controlling their posture beyond the resting position.” Also, we have already stated as a limitation that we need to verify the effectiveness of balance training with VR intervention for the elderly in the future.

Reviewer 2 Report

Overall, this study seems to contain interesting findings about balance exercise using VR. Although the manuscript is well structured, there are several areas that arouse my curiosity.

2.4. Statistical analysis

Why did the authors compare “Control” and “VR1 conditions”?

“ANOVA” should be modified to “Repeated measure ANOVA”.

  1. Results

For readership, it is recommended to visualize results as a graph including post-hoc analysis.

In Table 1, I cannot find any bolded font even though the authors explained that bolded font indicates significant difference.

Intuitively, I thought that VR3 and VR4 with high change rate of angle would be measured with high COP movement distance, or VR2 and VR4 with high tile angle would be measured with high COP. However, the experimental results showed that VR1 was the highest. It will be possible to increase the impact of the manuscript by accounting for reasons kindly.

Author Response

Response to Reviewer 2 Comments

Dear Editors:

Thank you for giving me the opportunity to submit a revised draft of my manuscript titled “The application of balance exercise using virtual reality for re-habilitation.” to Journal of Healthcare. We appreciate the time and effort that you and the reviewers have dedicated to providing your valuable feedback on my manuscript. We are grateful to the reviewers for their insightful comments on our paper. We have been able to incorporate changes to reflect most of the suggestions provided by the reviewers. We have highlighted the changes within the manuscript in yellow. Here is a point-by-point response to the reviewers’ comments and concerns indicated in red font.

General comments

Overall, this study seems to contain interesting findings about balance exercise using VR. Although the manuscript is well structured, there are several areas that arouse my curiosity.

2.4. Statistical analysis

Why did the authors compare “Control” and “VR1 conditions”?

Response: Thank you for your question. In this case, VR1 was set as the standard for the VR condition, so we first investigated whether there was any difference between the stationary state and the VR condition. We found that the amount of COP movement changed under the VR conditions, so we followed the steps of investigating which images were effective under different VR conditions, which is why we are applying these statistics to this case.

“ANOVA” should be modified to “Repeated measure ANOVA”.

Response: Thank you for pointing this out. Correction has been made. (p3, L101-102)

Results

For readership, it is recommended to visualize results as a graph including post-hoc analysis.

Response: Thank you for your valuable opinions. As you pointed out, we think it was difficult to understand, so we have also added the results of the posterior test. Please check it. (p4, L112-117-Table1)

In Table 1, I cannot find any bolded font even though the authors explained that bolded font indicates significant difference.

Response: Thank you for your suggestion. The p-values in Table 1 are in bold. Please check the following. (Table1)

Intuitively, I thought that VR3 and VR4 with high change rate of angle would be measured with high COP movement distance, or VR2 and VR4 with high tile angle would be measured with high COP. However, the experimental results showed that VR1 was the highest. It will be possible to increase the impact of the manuscript by accounting for reasons kindly.

Response: Thank you for your valuable feedback. Certainly, the larger the angle, the larger the COP movement can be expected. However, previous studies have reported that visual stimuli with a gravitational tilt of 20 degrees or more do not increase the perceived impact of visual stimuli, but rather attenuate it. Therefore, we believe that the present results are correct in that the COP movement increased the most at a 10-degree tilt. References [21] have been added to the discussion to elaborate on the explanation. (p4, L145-147)

Round 2

Reviewer 1 Report

The authors revised the manuscript titled “The application of balance exercise using virtual reality for rehabilitation”. They corrected errors and improved the manuscript. Also, they have incorporated changes to all of my suggestions. So, I accept the manuscript in its present form and recommend it to publish in the Journal of Healthcare.

Author Response

Thank you very much for your valuable time. We are happy that we were able to correct the points you pointed out. Thank you very much for reviewing my paper.

Reviewer 2 Report

Thanks for the detailed explanation.

All questions that I had are resolved. 

Author Response

(The authors gave the same response as above.)
